

# fLPS 2.0: rapid annotation of compositionally-biased regions in biological sequences

Paul M. Harrison

Department of Biology, McGill University, Montreal, QC, Canada

## ABSTRACT

Compositionally-biased (CB) regions in biological sequences are enriched for a subset of sequence residue types. These can be shorter regions with a concentrated bias (*i.e.*, those termed 'low-complexity'), or longer regions that have a compositional skew. These regions comprise a prominent class of the uncharacterized 'dark matter' of the protein universe. Here, I report the latest version of the fLPS package for the annotation of CB regions, which includes added consideration of DNA sequences, to label the eight possible biased regions of DNA. In this version, the user is now able to restrict analysis to a specified subset of residue types, and also to filter for previously annotated domains to enable detection of discontinuous CB regions. A 'thorough' option has been added which enables the labelling of subtler biases, typically made from a skew for several residue types. In the output, protein CB regions are now labelled with bias classes reflecting the physico-chemical character of the biasing residues. The fLPS 2.0 package is available from: https://github.com/pmharrison/flps2 or in a Supplemental File of this paper.

## INTRODUCTION

Biological sequences, despite being made from fixed alphabets of residues, demonstrate a wide diversity of sequence compositions. In particular, these sequences can be compositionally biased (CB) for a subset of the residue alphabet. For example, the protein sequence tract EDEEKDDELEIEEDEEDDDEDEDED is biased for E and D (glutamate and aspartate). If these tracts are sufficiently biased or repetitive over a short stretch, then they are termed 'low-complexity'. Also, one can have longer tracts that exhibit a milder compositional skew. In between, there are a continuum of CB cases (*Harrison, 2006*; *Harrison & Gerstein, 2003*). In proteins, CB regions are linked to distinct biophysical states such as intrinsic disorder, and to cell-structural proteins, fibrous proteins, and functional amyloids and prions (*Harbi & Harrison, 2014*; *Harrison, 2006*), and to the formation of intracellular biomolecular condensates or membraneless organelles (*Gomes & Shorter, 2019*). They also comprise part of the protein 'dark matter' that remains largely un- or under-characterized (*Harrison, 2018*); indeed, some CB dark matter is not assignable as

Corresponding author
Paul M. Harrison,
paul.harrison@mcgill.ca

intrinsically disordered or structured, and may give us clues to as yet unknown biophysical protein states (*Harrison, 2018*).

Several programs to annotate CB regions—and in particular, low-complexity (LC) regions—have been developed. These include SIMPLE (*Hancock & Armstrong, 1994*), SEG (*Wootton & Federhen, 1996*), CAST (*Promponas et al., 2000*), 0j.py (*Wise, 2001*), ScanCom (*Nandi et al., 2003*), CARD (*Shin & Kim, 2005*), BIAS (*Kuznetsov & Hwang, 2006*), LCD-Composer (*Cascarina et al., 2021*) and LPS / fLPS (*Harrison, 2006*; *Harrison, 2017*; *Harrison & Gerstein, 2003*). SEG annotates LC sequences by performing a scan using thresholds for sequence entropy and a fixed window length. It is used for masking LC sequences as part of the BLAST sequence alignment package (*Altschul et al., 1997*). Such masking is sometimes needed to avoid false inference of similarity by evolutionary descent (since these simpler sequences can arise independently multiple times during evolution quite easily). CAST annotates LC sequence by aligning to homopeptides of the twenty amino acids (*Promponas et al., 2000*). LCD-Composer uses a measure of amino-acid dispersion to characterise low complexity (*Cascarina et al., 2021*). The LCT web server analyzes the low-complexity and 'repeatability' of proteins sequences with a graphical output (*Mier & Andrade-Navarro, 2020*). Two other servers LCReXXXplorer and PLaToLoCo combine the results of multiple programs to graphically display LC regions (*Jarnot et al., 2020*; *Kirmitzoglou & Promponas, 2015*). The LPS algorithm used binomial probability to check for low-probability sequence regions, and was further developed into the fast algorithm fLPS, which can annotate the TrEMBL database in <1 h (*Harbi, Kumar & Harrison, 2011*; *Harrison, 2017*; *Harrison & Gerstein, 2003*). This algorithm has been applied successfully to the analysis of prions and prion-like proteins, and protein 'dark matter' (*An, Fitzpatrick & Harrison, 2016*; *An & Harrison, 2016*; *Harbi, Kumar & Harrison, 2011*; *Harrison et al., 2007*; *Harrison, 2018*; *Harrison, 2020*; *Su & Harrison, 2019*; *Su & Harrison, 2020*).

The fLPS program is especially useful for analyzing CB regions since: (i) it analyzes the full range of CB types (from low complexity to milder compositional skews); (ii) it characterizes both single- and multiple-residue biases; (iii) it does not require the specification of residue types by the user (although this option has now been added); (iv) it considers the differing background frequencies of individual residue types; (v) it is faster than the commonly used SEG algorithm (*Harrison, 2017*). fLPS has been applied to, for example, the identification of transactivation domains (*Arnold, 2018*), to analysis of the conservation of low-complexity regions in prokaryotes (*Ntountoumi, 2019*), analysis of low-complexity regions in stress granules (*Zhu, 2020*), and the delineation of domains in kinetochore proteins (*Cortes-Silva, 2020*) and in the PRR19 protein that functions in meiotic crossing over (*Bondarieva, 2020*), as well as in studies of prion-like protein evolution (*Harrison, 2020*; *An & Harrison, 2016*; *An, Fitzpatrick & Harrison, 2016*; *Morgulis et al., 2006*; *Wong, Maurer-Stroh & Eisenhaber, 2010*). Here, the latest fLPS 2.0 package is reported. In this package, the program flow has been modified to add consideration of DNA sequences; also, the user can specify subsets of residue types and existing domain annotations to filter from sequences, in order to discover discontinuous biased regions. The baseline precision of the algorithm can be adjusted to discover more mildly biased regions that may have biological significance. Examples of fLPS 2.0 application are presented and discussed.

## METHODS

### Implementation

The fLPS 2.0 package is written in standard C. The name 'fLPS' stands for fast LPS, where LPS stands for 'Low Probability Subsequences'. The package comprises the source code and executables compiled for MacOSX and Linux. There is the fLPS program itself, plus two accessory programs: *CompositionMaker*, which can be used to calculate background residue compositions; and *DomainFilter*, which is used to either excise or mask previously annotated domains (such as those with known protein structure, see section immediately below). Each of the programs works on input files of any size in standard FASTA format. The package is available at https://github.com/pmharrison/flps2 and in File S1.

### Algorithm and new added features

The program fLPS works through a process of binomial probability (*P*-value) minimization, as described in detail previously (*Harrison, 2017*). There are four main steps that are summarized in Fig. 1 at the top of the figure: *(i) QUICK SCAN; (ii) MINIMIZE; (iii) MERGE; (iv) OUTPUT*. At the end of the process, single-, and multiple-residue LPSs, are output if they are below the user-specified *P*-value threshold, or default threshold. Biased regions are labelled with a *bias signature* which is a list of the biasing residues in order of bias precedence delimited with curly brackets. At each of these stages, efficiency measures are taken to avoid or delay probability calculations unless/until they are necessary (*Harrison, 2017*).

The following are the main new options added to the fLPS code:

(i) *Precision of the calculation:* In the initial *QUICK SCAN* step, by default (the '*–z fast*' option), windows with a *P*-value below the baseline threshold of 0.001 are considered. Also, the windowing along the sequence proceeds with a step size = 3 residues (Fig. 1). This means that some regions that are made from biases for a larger number of residue types might be missed; also, short regions with a milder bias that might have biological significance could sometimes be overlooked. Therefore, options for the base-line precision of the program have been added. If '*–z medium*' is specified, the base-line *P*-value threshold is set to 0.01, with a windowing step size = 2. For the most precise option '*–z thorough*', the base-line *P*-value is 0.1 and the step size = 1 (Fig. 1). However, these latter two options can produce a huge amount of output for larger databases, so they should be applied to such databases with caution.

(ii) *DNA analysis:* DNA sequences can be specified using the *–n* option. By default, each of the four bases A, G, C and T has equal background probability.

(iii) *Domain filtering:* Using the *DomainFilter* accessory program, previously annotated domains can be filtered in either of two ways, *i.e.,* either 'excised' or 'masked'. When 'excised' is specified, *DomainFilter* outputs shorter sequences, with the domain sequences removed. The 'masked' option outputs sequences with the domains masked with Xs. The positions of the excised or masked domains are labelled on the name line of the sequences in the FASTA-format file. When the FASTA-format output file from *DomainFilter* is used as input for fLPS bias annotations, the domain positions appear in the fLPS output if the ''-option *–D* is specified.
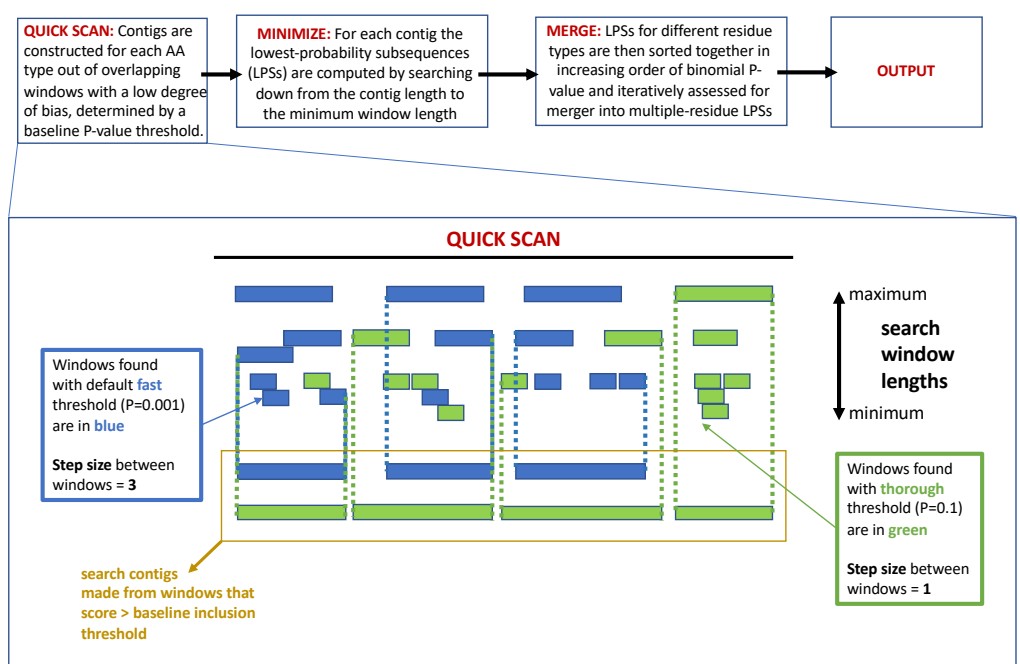

**Figure 1** **A schematic detailing the –z option to adjust the base-line precision of the fLPS calculation.** At the top of the figure is a pipeline summarizing the basic fLPS algorithm. Below that is detailed the effect of the –z option for adjusting the base-line precision of the algorithm. In the *QUICK SCAN* stage, when the –z *thorough* option is specified, more windows are stored in accord with the higher base-line *P*-value (these are coloured green). Thus, there are more and longer search contigs (surrounded with yellow box) at the end of this stage.

(iv) *Restriction lists:* With the –r option, the user can specify a subset of residue types, *e.g.*, only negatively-charged amino acids (E, D), or the six-membered aromatics (F, Y, W).

Further option additions include: An option $(-O)$ option to specify a prefix for a unique output filename that also contains the parameters used in running the fLPS program; a '–o oneline' output option, wherein the results for each sequence are listed in a single-line summary; a –k option to ignore the unknown residues in calculations ('X' for proteins and 'N' for DNA). The output has also been updated to include further new features. A calculation of the enrichment of the biasing residues in the output LPSs has been added, which is the proportion of biasing residues in the LPS divided by the total expected background frequencies of the biasing residues. To enable quicker characterization of bias trends in a data set, '*bias class*' labels are now featured for both protein and DNA sequences. For proteins, these labels are derived from the Taylor amino-acid classification Venn diagram, with some additional categories (*Taylor, 1986*). The applicable class label that has the smallest membership is picked, when assigning these. For DNA, these labels represent the eight possible compositional biases: {A}|{T}, {G}|{C}, {AT}, {GC}, {AC}|{GT}, {AG}|{CT}, {ATC}|{ATG} and {ACG}|{CGT} (discussed below).

For better annotation of short low-complexity regions, trimming of LPSs of minimum window length is now employed. That is, if possible, residues are sheared off both ends

of the minimum-length LPS if they do not contribute to the bias. This improves the annotation of ~6–8% of LPSs, in trials on the *S. cerevisiae* S288C proteome (downloaded from UniProt reference proteomes (*Boeckmann et al., 2003*)) using a variety of parameters.

### Example data

The UniProt canonical reference human and budding yeast (*S. cerevisiae* strain 288C) proteomes were downloaded from http://www.uniprot.org in January 2021 (*Boeckmann et al., 2003*). The human proteome was cross-referenced with the InterPro list of domain annotations downloaded from https://www.ebi.ac.uk/interpro/, to make a list of human proteins that contain the RRM RNA-binding domain (*Blum et al., 2021*).

Human promoter data was obtained from the EPD eukaryotic promoter database (*Schmid et al., 2004*). These were a set of representative promoters (one per gene) defined by the EPD. Sequences spanning from −999 to +100 around the transcription start site were analysed.

### Prion-like regions

Prion-like regions were annotated for the human proteome using the PLAAC program with default parameters (*Lancaster et al., 2014*).

## RESULTS

### Using the new domain-filtering and restriction list options of fLPS: application to analysis of human RNA-binding proteins

It is often advantageous to restrict CB annotation to a subset of residues to enable easy counting of different types of bias region. Users are now able to restrict their bias annotation using a 'restriction list' specified with the –*r* option of fLPS (Fig. 2A). Also, it is possible that certain proteins have discontinuous CB regions, *i.e.,* the CB regions may have small, structured domains embedded in them, or they may be comprised of the loop regions within a single protein domain. To enable discovery of such discontinuous CB regions, the *DomainFilter* program can be used to excise or mask domains or domain parts before using the fLPS program (Fig. 2A). These two options were combined in analyzing the CB regions of human RNA-binding proteins, specifically those containing the RRM RNA-binding domain (Fig. 2). The RRM domain is used in eukaryotes to bind RNA during diverse cellular processes, and is typically associated with intrinsically-disordered regions (*Su & Harrison, 2020*). After applying *DomainFilter* to excise Pfam protein domain annotations (*Mistry et al., 2021*), the main single-residue biases were assessed with an initial run of fLPS (those having > 50 cases); thereafter, a final run of fLPS used a restriction list based on these main single-residue biases to enable better counting of bias types.

There are 206 RRM-domain-containing human proteins in this protein data set. The most common multiple- and single-residue biases involve arginine, serine, proline and glycine, and are associated mostly with 'mixed', 'polar', 'small' and 'charged' bias classes; glutamine and asparagine biases, which are associated typically with prion-like domains, are only of middling abundance (Figs. 2B–2D). Indeed, although prion-like domains are often cited as being associated with RNA-binding proteins, in this case they only occur

Peer J

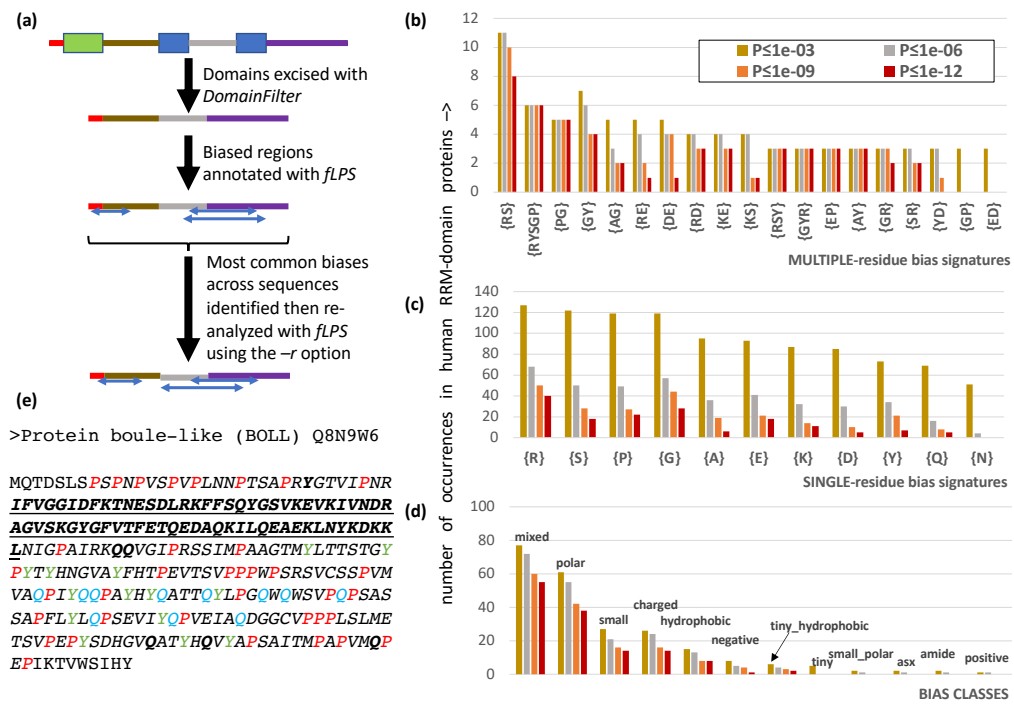

**(e)**
```
>Protein boule-like (BOLL) Q8N9W6

MQTDSLS PSPNPVSPVPLNNPTSAPRYGTVIPNR
IFVGGIDFKTNESDLRKFFSQYGSVKEVKIVNDR
AGVSKGYGFVTFETQEDAQKILQEAEKLNYKDKK
LNIGPAIRKQQVGIPRSSIMPAAGTMYLTTSTGY
PYTYHNGVAYFHTPEVTSVPPPWPSRSVCSSPVM
VAQPIYQQPAYHYQATTQYLPGQWSVPQPSAS
SAPFLYLQPSEVIYQPVEIAQDGGCVPPPLSLME
TSVPEPYSDHGVQATYHQVYAPSAITMPAPVMQP
EPIKTVWSIHY
```

**Figure 2** **Analysis of RRM RNA-binding domain proteins in the human proteome.** (A) Human Pfam domain annotations (coloured boxes) are excised with *DomainFilter* and the biases are annotated using fLPS 2.0 with human proteome background composition. The most prevalent single-residue biases (occurring > 50 times) were picked out (listed in part (C)) and used as a restriction list with the –*r* option. (B) A bar chart of the most prevalent multiple-residue bias signatures (that occur for any threshold ≥ 3 times). The data for four *P*-value thresholds are shown. (C) As in (B) except that single-residue biases are counted. (D) As in (B), except the bias classes are counted. The following bias classes do not occur: glx, tiny_polar, polar_aromatic, aliphatic, aromatic. (E) An example of a discontinuous biased region from human BOLL protein. The RRM domain (Pfam PF00076, underlined bold) is excised. A {P} CB region with *P*-value = 7.2e−9 is shown in *italics* with the P residues in red. There are also a {Y} CB region (*P* = 8.6e−6, residues in green) and a {Q} CB region (*P* = 2.8e−5, residues in blue). These go together to make a {PYQ} region of the same extent as the {P} region with *P* = 4.4e−13. Other Q and Y residues in this multiple-residue CB region are in bold.

in ∼1 in 6 RRM-containing proteins, as judged by the PLAAC program (*Lancaster et al., 2014*) (Fig. S1, 32/206 (15.5%) have PLAAC LLR scores ≥ 15.0, and 35/206 (17.0%) have PLAAC PRD scores > 15.0). These PLAAC prion-like regions arise despite only moderate asparagine and glutamine frequencies and are thus substantially dependent on other residues that are common in prion-forming domains, such as tyrosine, glycine and serine, which are common biases in the RRM-containing proteins (Figs. 2B–2D). In Fig. 2E, the human BOLL 'protein boule-like' is presented as an example of a discontinuous CB domain around an RRM domain.

## Increased precision with the –z option

As described in *Methods*, the –*z* option can be used to increase the precision of the initial scanning by the fLPS algorithm for compositional deviations (Fig. 1). Two examples

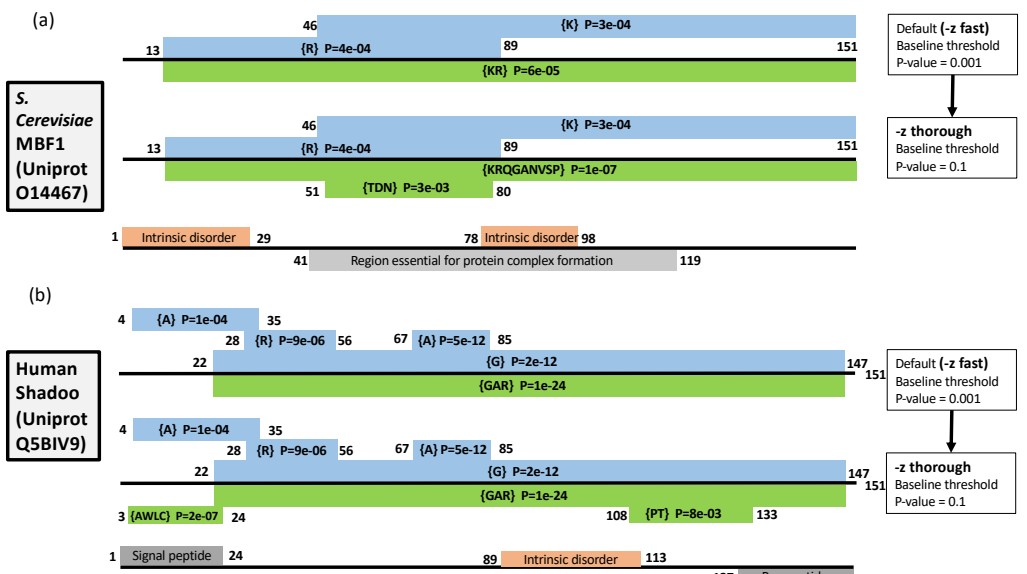

**Figure 3** Two examples of the effect of the –z option: (A) Multiprotein-bridging factor MBF1 from *S. cerevisiae*; (B) human Shadoo protein. Single-residue CB regions are depicted as blue boxes and multiple-residue as green. They are labelled with their biases and binomial *P*-values, and their endpoints. Intrinsic disorder and other domain annotations are labelled in orange and grey respectively (and are taken from UniProt (*Boeckmann et al., 2003*). At the top of each panel are depicted the annotation from the default –z fast option, below that the annotations after using the –z thorough option, then at the bottom of each panel are the UniProt sequence annotations.

of the effects of this option are illustrated (Fig. 3). Multi-protein bridging factor MBF1 is a transcriptional coactivator that promotes GCN4-dependent transcriptional activity by bridging between the DNA-binding areas of GCN4 and TATA-binding protein. The default fLPS settings detect a mild bias for positively-charged residues *{KR}*, which becomes a stronger bias comprised of further biasing residues *{KRQGANVSP}* when the 'thorough' option is applied. Also, a region weakly biased for polar residues *{TDN}* appears (Fig. 3A). These biases are likely linked to DNA and protein interactions within complexes. The second example is the Shadoo protein from human (Fig. 3B). Shadoo is a member of the prion-protein (PrP) family that has demonstrated some neuroprotective behaviour (*Westaway et al., 2011*). Like PrP, the protein that underlies prion diseases, it contains CB and intrinsically-disordered regions. Here, the major CB annotations are stable when –z thorough is specified, but additional mildly biased regions are detected, one of which corresponds to a signal peptide, the other an area bridging between and intrinsically disordered region and a pro-peptide (Fig. 3B).

These examples demonstrate three effects of increasing the precision of the initial compositional scanning: *(i)* mildly biased tracts are detected that can be quite short and that may have biological significance; *(ii)* further bias detail is sometimes added to CB regions, decreasing the binomial *P*-value; *(iii)* tracts with a bias made from several residues and that were previously not detectable (such as the {AWLC} tract in Fig. 1B) become

**Table 1** Comparison of results for the precision options, using the yeast proteome as input[*].

| P-value thresholds Precision option (–z) ↓ | Number of single-residue CB regions | | | Number of multiple-residue CB regions | | |
|---|---|---|---|---|---|---|
| | $P \leq 1e-03$ | $P \leq 1e-06$ | $P \leq 1e-09$ | $P \leq 1e-03$ | $P \leq 1e-06$ | $P \leq 1e-09$ |
| Fast (default) | 32022 | 5781 | 2268 | 6336 | 4512 | 2744 |
| Medium | 36589 | 5792 | 2275 | 17117 | 6350 | 3395 |
| Thorough | 37738 | 5792 | 2276 | 27229 | 7675 | 3766 |

Notes.

[*]UniProt reference proteome for *S. cerevisiae* 288C, downloaded January 2021.

**Table 2** Number of *S. cerevisiae* proteins with signal peptides that coincide with CB regions annotated by fLPS[*].

| P-value thresholds Precision option (–z) ↓ | $P \leq 1e-03$ | $P \leq 1e-04$ | $P \leq 1e-05$ |
|---|---|---|---|
| Fast (default) | 60/301 (19.9%)[**] | 25/301 (8.3%) | 15/301 (5.0%) |
| Medium | 111/301 (36.9%) | 51/301 (16.9%) | 29/301 (9.6%) |
| Thorough | 180/301 (59.8%) | 90/301 (29.9%) | 48/301 (15.9%) |

Notes.

[*]Annotations for signal peptides were taken from UniProt (301 in total).

[**]The numbers of signal peptides for which ≥50% of their residues overlap ≥50% of the residues of an individual fLPS-annotated CB region.

evident. In aggregate, these three effects increase the ability of the program to delineate compositionally-defined domains in proteins. As shown in Table 1 detailing analysis of the *S. cerevisiae* proteome, a significant number of further multiple-residue CB regions are detected, even for smaller *P*-value thresholds (such as $P \leq 1e-09$). Since one of the tracts in Fig. 1B corresponds to a signal peptide, the correspondence between signal peptide positions in the *S. cerevisiae* proteome and CB regions was also examined (Table 2). The number of signal peptides corresponding to CB regions increases to a highest value of ~60% with a $P \leq 1e-03$ bias *P*-value threshold. The results are generally in line with a previous analysis of sequence complexity in signal peptides, where 24% of residues of signal peptides in analyzed data sets were labelled part of low-complexity tracts by SEG (*Wong, 2010*; *Wootton, 1996*).

## Analysis of DNA sequence

DNA sequences can be analyzed by specifying the –*n* option; by default, each base is expected with equal background probability. In total, there are 40 different possible biases (Fig. 4A). These can be segregated into eight bias classes for DNA (Fig. 4A). The last two of these bias classes, {ATC}|{ATG} and {ACG}|{CGT}, correspond to strand-specific depletions of single bases, *i.e.,* {ATC}|{ATG} indicates a strand-specific lack of C or G. An example of a DNA CB region from a human promoter is illustrated (Fig. 4B). To illustrate its application to DNA, we used the fLPS program to examine bias trends in a representative data set of human promoters taken from the EPD database (Fig. 4C). Interestingly, the

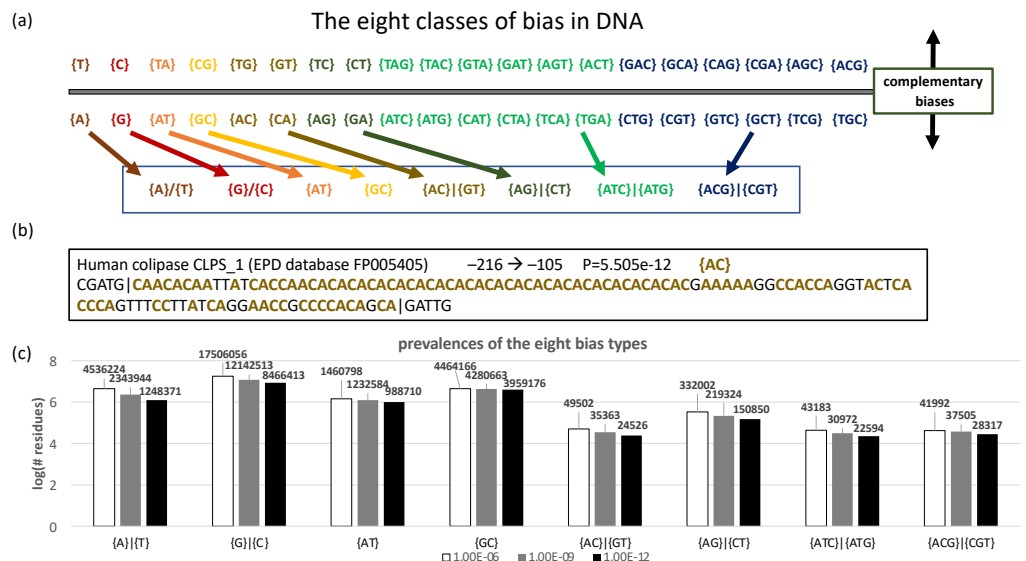

**Figure 4** Analysis of DNA. (A) Eight classes of bias are possible in DNA. Complementary biases are arrayed above and below the line, *i.e.,* a bias on one strand for {*GT*} (guanine and thymidine) corresponds to a bias for {*CA*} (cytidine and adenine) on the complementary strand. Biases with the same colour are summarized with one of the eight basic bias class labels (in the box at the bottom of the panel). (B) An example of a biased region in human promoter DNA, for colipase CLPS_1. The position in the promoter (downloaded from the EPD database *Schmid et al., 2004*) is indicated, along with the bias signature *AC*.

tri-base {*ATC*}|{*ATG*} and {*ACG*}|{*CGT*} bias classes are almost as prevalent as the two-base {*AC*}|{*GT*} bias class.

## DISCUSSION

The known protein universe contains much 'dark matter', some structured, some intrinsically disordered, some not assignable as either (*Harrison, 2018*). The present fLPS package helps to address one aspect of this protein dark matter, which is that it often has unusual amino-acid composition, the structural properties of which have yet to be characterized. As examined above for human RRM-domain-containing proteins, some of this dark matter may be discontinuous CB regions that have smaller structured domains embedded in them. Such discontinuity might also be possible within a protein domain, *e.g.,* in the loops of a transmembrane domain. fLPS can also be used to assess whether such compositional biases are unusual relative to the background proteome composition of a particular organism or clade, or are part of organism- / clade-specific trend. CB domains, such as the proline-rich region in PRR19 protein {*Bondarieva, 2020* # 37} that functions in meiotic crossing-over, may have specific functional relevance. Parsing proteins into subdomains, including milder CB domains found with an increased baseline thoroughness (the −z option), may help in the generation of experimental constructs and further hypotheses for experiments. They are also useful for studying proteome-wide trends to gain more general functional or evolutionary insights (*e.g.,* refs. {Ntountoumi, 2019 #34}{Su, 2020 #38}). Varying the parameters (in particular −*m*, −*M*, −*t*, and the new

parameters *–r* and *–z*) can help to delineate possible biologically meaningful subdomains in larger biased tracts or within intrinsically disordered regions, such as in prion-forming proteins {*Harrison, 2017* #22}.

The fLPS package program flow was modified to accommodate the option of analyzing DNA sequences. I applied this option to a set of representative human promoters, as an example. Beyond the standard conception of DNA bias as either {GC} or {AT}, substantial tracts of other possible biases were observed, including strand-specific dearths of single bases (*i.e.,* the bias classes {ATC}|{ATG} and {ACG}|{CGT}). It would be interesting to investigate experimentally whether such DNA CB domains have a general biological significance. To my knowledge, there is not a currently available program that delineates all of the possible biased domains of DNA in this way (other programs, such as Dustmasker (*Morgulis, 2006*) or TANTAN (*Frith, 2011*), are designed to tackle the problem of avoiding spurious alignments, which is not what fLPS is designed for.)

Intrinsically disordered regions (IDRs) in proteins were initially discovered as long stretches of amino acids in proteins that remain unfolded under physiological conditions [1, 2] (*Uversky, 2002*). Compositional bias or 'low complexity' is a characteristic feature of intrinsically disordered regions (IDRs), although there is substantial overlap in sequence complexity values between IDRs and ordered regions (*Pedro Romero, 2001*). Also, different definitions of sequence complexity or compositional bias have different degrees of linkage to disorder or order, with tandem-repeat tracts more likely to encode ordered regions (*Mier, 2020*). Because of this link, fLPS 2.0 may be useful for the characterization of subdomains in intrinsically-disordered proteins. The boundaries of compositionally-defined domains may differ to those of IDRs, IDRs may be split into multiple compositionally-defined regions, or new algorithmic scenarios using the definition of compositionally defined domains may enable the annotation of further intrinsic disorder (*Necci, 2021*; *Sirota, 2010*; *Tang, 2021*).

## Examples of running the package

A diverse choice of parameters is possible in running the fLPS 2.0 program. Here are some examples:

(1) *Annotating low-complexity regions:* For the specific task of annotating short CB regions of the sort termed 'low-complexity', the following parameters are suitable (with the yeast proteome 'yeast.fasta' as an example input file):

*./fLPS –t1e−5 (or –t1e−6) –m5 –M25 –o long yeast.fasta*

(2) *Analyzing for discontinuous CB domains:* Firstly, structured domains are excised from the sequences, then fLPS is run using the –D option:

*./DomainFilter –D excised yeast.fasta >yeast.Dexcised.fasta*
*./fLPS ''—D yeast.Dexcised.fasta*

(3) *Restricting biases to specific sets:* To analyze biases for just six-membered aromatic side-chain amino acids only (F, Y and W), using the yeast proteome background composition:

*./CompositionMaker yeast.fasta (makes file 'yeast.fasta.COMPOSITION')*
*./fLPS –dv –ooneline –c yeast.fasta.COMPOSITION –r FYW yeast.fasta*

Also specified are headers and footers (*–d*), oneline output format (*–o oneline*) and verbose behaviour during runtime (*–v*).

(4) *Annotating longer biased regions with the thorough option:* To find longer biased regions that have compositional skew the following options may be suitable:

*./fLPS –z thorough –t0.001 –M 1000 yeast.fasta*

(5) *DNA:* For DNA, the "—n option is specified:

*./fLPS –dn DNA.example.fasta*

Here, headers and footers are also output (*–d* option).

## CONCLUSIONS

The fLPS 2.0 package is a versatile package for annotating compositional biases, either 'low-complexity' regions, or regions with milder or long-range compositional skew.

Users can now apply the package to DNA to identify all the possible DNA CB domains. In addition to the unique features of fLPS listed at the end of the Introduction, utility is gained from the added domain filtering, restriction list and precision options, which can be combined to identify CB domains in support of experimental hypotheses. The package is available from: https://github.com/pmharrison/flps2 or File S1.

### Funding
This work was funded by a Discovery Grant from the Natural Sciences and Engineering Research Council of Canada. The funders had no role in study design, data collection and analysis, decision to publish, or preparation of the manuscript.

### Grant Disclosures
The following grant information was disclosed by the author:
The Natural Sciences and Engineering Research Council of Canada.

### Competing Interests
The author declares there are no competing interests.

### Author Contributions
- Paul M. Harrison conceived and designed the experiments, performed the experiments, analyzed the data, prepared figures and/or tables, authored or reviewed drafts of the paper, wrote program code, and approved the final draft.

### Data Availability
The code is a TAR archive of the code for the package available in the Supplemental File. It also includes executables for Linux and MacOSX operating systems.
The code is also available at GitHub: https://github.com/pmharrison/flps2.

## Supplemental Information

Supplemental information for this article can be found online at http://dx.doi.org/10.7717/peerj.12363#supplemental-information.

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
