# Peer review of "fLPS 2.0: rapid annotation of compositionally-biased regions in biological sequences"

_PeerJ, doi:10.7717/peerj.12363_

## Round 0.1 · original submission · Major Revisions

Please address issues pointed by both reviewers and revise your manuscript accordingly.

Reviewer 1 ·

Basic reporting

1. Overall, clearly written, nice figures.

2. The program fLPS is updated, but this term itself was not defined clearly. What do the letters mean- I assume "fast low-probability sequence regions"?

3. Lines 70-81. It is not fully clear why this new version of fLPS is better or an advance over the prior version, or with respect to other programs. Are the updates enough to warrant a new publication? Explain more clearly why these updates are essential, and why they warrant a new publication.

Experimental design

Lines 94-111 appears to be repeating the program methods from the author's 2017 paper. Could be much shorter, and emphasize that this is not new to the present paper.

Lines 114-143 provides a description of the updates in the fLPS program. Some of these updates appear very minor such as providing an output file name option "-O". Or a "restriction list" with a "-r" option. Again, please make more clear why these updates are considered major and important enough to warrant a new paper.

Lines 229-232. Similarly, is this enough benefit to justify publication of the update? Same for lines 234-244. It may be enough, but there should be a better explanation of why.

Validity of the findings

The findings appear valid. The author should improve explaining why researchers would find this program useful for them. The patterns may exist (e.g. Fig. 4), but why are they important, or why should one be interested in studying those patterns found using the updated program.

I am wondering why the author does not provide a web server, or user friendly version particularly for researchers interested in this analysis, that may prefer a more user-friendly version, and better visualization (graphical display) of results. This may be a substantial drawback, as some biologists would like to conduct such analyses, but in a more user-friendly way.

Additional comments

Overall, the manuscript is well written and clear. I suggest improvements in explaining why this update should be considered crucial and worth a new publication over the original, and why researchers should be interested in the outputs and patterns detected only from the update. Also, why a user-friendly version appears to not be available (if it is, then the author needs to state this clearly). I suggest the manuscript would be improved by addressing these points, and can be published at PeerJ.
This is between a major and minor revision. I will assume the author can address these points and thus selected minor.

Reviewer 2 ·

Basic reporting

The proposed MS is largely a technical text decribing update details of the fLPS software and their impact on application of fLPS in sequence analysis and annotation efforts. At the same time, it is more than a software application note as it dives into review aspects such as the definition of sequence bias/low complexity/etc., the impact for biological function, comparative mentioning of some of the most used software tools and of a few major insights achieved with sequence bias studies. This rather enhances the value of the text that, generally, provides a smooth, mature impression and seems to be correct in its critical statements. The application examples seem useful.

Experimental design

n/a

Validity of the findings

There are a few questions that remain in the mind of the reader such as (1) how does fLPS fair with problematic sequence elements such as TM regions or signal leader peptides that sometimes pop up as disordered/low complexity (PMID 20686689, 24890864) or (2) can fLPS recreate the list of regions in some databases for these specific sequences types (e.g., PMID 27899601).

Additional comments

The author can be proud of his contributions to the field; yet, the reviewer feels that the author goes overboard in certain aspects.

The statement “The fLPS program is optimal for analyzing CB regions since ...” (line 70) is a no-go and should be toned down, for example by “The fLPS program is especially useful for analyzing CB regions when ... the following aspects are at the center of attention:”.

The author has managed to include all of his previous papers into the reference list, even those with (though justifiable) distant relationship to the matter. This is fully appropriate if ... but, essentially, no other literature is mentioned except if technically inavoidable. It appears that the author does not exactly follow the publications in the field. Otherwise, he would have noticed that the Section Editor of this journal has a history in sequence disorder (e,g, PMID 11784292, a review). It is not the reviewer’s attempt to provide a full list here, just a few pointers: recent reviews (PMID 30698641, 33875885), a method to make the disorder definitions from several tools more comparable (PMID 20158872), other methods (PMID 32702119, 32535960) or a meta-server (PMID 32421769). There are databases of disordered regions/IDPs (e.g., PMID 29385418), etc. The article would benefit from putting the updated software tool into the real context of the state of the art.

---

## Round 0.2 · accepted · Accept

Thank you very much for addressing the critiques of the reviewers and for the careful revision of your manuscript. I am pleased to accept it now.

Reviewer 2 ·

Basic reporting

The MS is a straightforward, pleasant reading. The revisions have further improved the MS.

Experimental design

no critical comment

Validity of the findings

no critical comment

Additional comments

n/a